# A Pilot-Scale Nanofiltration–Ultrafiltration Integrated System for Advanced Drinking Water Treatment: Process Performance and Economic Analysis

Fengxia Chen [1,2], Lifang Zhu [2,*], Jianzhong Tang [3], Dongfeng Li [2], Fang Yu [3], Fuqing Bai [2], Zhou Ye [1], Lu Cao [1] and Nan Geng [2]

[1] School of Marine Engineering Equipment, Zhejiang Ocean University, Zhoushan 316022, China; fengxia5206@163.com (F.C.)
[2] College of Water Conservancy and Environmental Engineering, Zhejiang University of Water Resources and Electric Power, Hangzhou 310018, China; baifq@zjweu.edu.cn (F.B.)
[3] Haiyan Enterprises Water Group Limited, Jiaxing 314300, China; tjz660760@163.com (J.T.); 13625836768@163.com (F.Y.)
* Correspondence: zhulf@zjweu.edu.cn

**Abstract:** In this pilot study, the performance of an "ultrafiltration (UF) + nanofiltration (NF)" advanced treatment process in improving drinking water quality was investigated. The membrane performance and effluent qualities of three commercial NF membranes (Dow Filmtec NF270-400, VONTRON TAPU-LS, and GE Osmonics-HL8040F 400) were evaluated, and the reasons for the difference in effluent quality of these three NF membranes were analyzed. The results showed that UF as a pretreatment process could provide NF with stable and qualified influent. After passing through the UF unit, the turbidity of raw water decreased by 88.6%, and the SDI value was less than 3. Due to the small pore size of NF membranes, organics and polyvalent ions in raw water were further removed. With a water recovery of 90%, the conductivity, chemical oxygen demand ($COD_{Mn}$), and hardness of NF effluent are significantly improved. The three commercial NF membranes showed different performance advantages. Among them, Dow Filmtec NF270-400 had the best desalting performance, VONTRON TAPU-LS had the highest retention rate of organic matter, and GE Osmonics-HL8040F 400 had significantly advanced softening performance. Thanks to the combination of the UF membrane and NF membrane, membrane fouling was effectively inhibited, and drug consumption was within an acceptable range. The operation costs of these three NF membranes were 0.165, 0.179, and 0.171 USD per ton of produced water, respectively. The results showed that the UF + NF process is an ideal technology for advanced treatment in water plants.

**Keywords:** drinking water; advanced treatment; ultrafiltration; nanofiltration

## 1. Introduction

With the continuous improvement of urbanization and industrialization in China, water pollution has gradually intensified, and the main contradiction in the water supply industry has shifted from water shortage to water quality [1,2]. The quality and quantity of drinking water are directly related to the vital interests of human beings [3]. Quantity and quality priority has become the consensus of water industry researchers and local water authorities [4]. At present, more than 95% of water plants in China still use conventional treatment processes when dealing with microbial and organic pollutants, including coagulation, sedimentation, filtration, disinfection, etc. [5,6]. Although the treated water meets the current national drinking water sanitation standards, disinfection byproducts (DBPs) and trace organic pollutants are important obstacles to the improvement of drinking water quality [7–9].

For organic pollutants and DBPs precursors that cannot be effectively removed by conventional treatment, advanced drinking water treatment is usually needed to improve

the water quality assurance rate. Advanced water treatment technology mainly includes ozone–biological activated carbon technology, membrane separation technology, and adsorption [10,11]. Ozone–biological activated carbon technology can strengthen the removal of organic pollutants, $NH_3$-N, etc., and has been applied in many water plants. However, there are still some problems in practice, such as the formation of bromate byproducts and the limited removal rate of some trace organics [12,13]. Moreover, with the saturation of activated carbon adsorption, the late processing capacity is difficult to guarantee. Membrane separation technology is a pure physical process which greatly reduces the amount of disinfectant, solves the problem of chlorination disinfection byproducts to a certain extent, and ensures the chemical safety of drinking water more significantly [14–16]. The use of adsorbents to remove transition metal ions and organic matter from water is also a very effective method. Recently, microgels have attracted a lot of attention because of their prospective applications in water treatment [17]. Arif et al., (2023) reported that S@P(NVCL-AA) microgel, as an adsorbent, was effective in removing iron (III) ions and organic matter from water [18].

The membranes commonly used in water treatment mainly include microfiltration (MF), ultrafiltration (UF), nanofiltration (NF), and reverse osmosis (RO) [19]. Nevertheless, the separation process of different membranes is not the same in terms of material passage or retention. The UF membrane is a kind of porous membrane with large pore size, which is 1~20 nm. Its separation mechanism is the "pore size sieving effect", that is, molecules smaller than the pore size can pass through, and molecules larger than the pore size are retained by the membrane [20,21]. Therefore, macromolecules such as suspended matter, colloid, particles, bacteria and viruses in water can be trapped by UF membranes. Li et al. (2022) reported that the ceramic UF membranes could reduce the water turbidity from ~1.5 NTU to <0.1 NTU, while the removal efficiency for organic matter was limited [22]. An RO membrane is a kind of dense membrane based on the "dissolution–diffusion" separation mechanism, according to the different dissolution rates and diffusion rates of substances, to achieve the separation of different substances [23,24]. The pore size of RO membranes is generally between 0.1~0.7 nm, so it can effectively achieve deionization and remove organic matter [25]. Because of this, the operating pressure of RO membranes is significantly higher than that of UF membranes and NF membranes. NF membranes have nano-scale pore sizes (1~2 nm), and their molecular weight cutoff (MWCO) is between that of UF and RO membranes. In addition, the surface of NF membranes usually carries a certain charge, and its separation mechanism also needs to consider the influence of surface potential. Previous studies have shown that the retention of electrically neutral substances by NF membranes is mainly controlled by the mechanism of "pore size sieving" [26–28]. Furthermore, due to the large interaction force between polar substances and nano-sized pores, the polarity of the substances also has a certain impact on the interception performance [29]. The retention of inorganic salts such as $Ca^{2+}$ and $Mg^{2+}$ by NF membranes is also affected by potential

Despite the fact that NF membranes can effectively remove small molecule organic matter and inorganic salts dissolved in water, the strong adsorption capacity of NF membranes makes them become a hotbed of microorganisms [30,31]. Once biofilm is formed on the membrane surface, it is very difficult to clean up, even through chemical cleaning [32]. Without proper pretreatment, the life and treatment effect of NF membranes will be greatly reduced. Compared with NF membranes, UF membranes have a natural advantage in resisting membrane fouling due to their larger pore size, and have lower operating pressure and energy consumption. With the increasing demand for water quality and safety, a UF + NF process combination is expected to be the preferred choice for advanced treatment in water plants.

To date, most of the research on NF applications is focused on the short-term and laboratory scale. Little information is available on pilot-scale studies of advanced treatment of actual drinking water using a UF + NF process. The purpose of this pilot study is to investigate the process performance of a UF + NF system in the advanced treatment of drinking water, and to dissect the relationship between effluent quality and membrane

properties. The results will provide technical support for the process selection of water plants and provide a reference for the popularization and application of NF advanced treatment technology for drinking water.

## 2. Materials and Methods

### 2.1. Raw Water

#### 2.1.1. Water Plant Location

As one of the most economically developed regions in China, Taihu Lake basin is a typical plain river network area with a dense population and developed industry. Due to the lack of quality water sources, the water supply in this region is dependent on advanced treatment from water plants. With the increasing demand of residents for tap water quality, the advanced drinking water treatment processes of local water plants are aiming to upgrade. The pilot-scale study was conducted in a water plant located in South Taihu Lake Basin.

#### 2.1.2. Raw Water Quality

The details of water source quality can be found in our previous work [33]. At present, the conventional drinking water treatment process of "coagulation + sedimentation + filtration + chlorine disinfection" is adopted in this water plant. The raw water in this pilot study was the effluent from the activated carbon filter. The water quality of the source and the effluent of this water plant under current working conditions are listed in Table 1. The common contaminants in water sources are $COD_{Mn}$, $NH_3$-N, nitrate, chloride, sulfate, suspended solids, and total bacterial count, etc.

**Table 1.** The water quality of source and effluent.

| Parameters | Water Source [1] | Effluent [2] |
|---|---|---|
| pH | 7.71~8.13 | 7.40~7.55 |
| Turbidity (NTU) | 14.41~43.15 | 0.072~0.300 |
| Total Hardness (mg/L) | - | 144.13~189.17 |
| $COD_{Mn}$ (mg/L) | 3.70~5.40 | 1.26~2.24 |
| $NH_3$-N (mg/L) | 0.10~0.40 | <0.02~0.35 |
| Nitrate (mg/L) | 0.21~4.02 | 1.79~2.80 |
| Chloride (mg/L) | 34~70 | 72~85 |
| Sulfate (mg/L) | 46~61 | 61~71 |

[1] Water source quality was monitored monthly from January to December 2022. [2] Effluent quality refers to the effluent from the activated carbon filter.

### 2.2. Pilot-Scale Process

In this pilot study, a UF + NF process was used to improve effluent quality. The improvement effects of different types of NF membranes produced by different manufacturers were compared by monitoring pH, turbidity, $COD_{Mn}$, $NH_3$-N and other parameters. The results will provide a reference for water plants to evaluate the suitability of the UF + NF process and to select membranes. The specific process is shown in Figure 1.

The UF + NF integrated system is composed of a UF unit and an NF unit. The raw water came from the activated carbon filtration tank of the water plant, and was pumped into the UF unit at a flow rate of 5.2 m$^3$/h. The UF unit performed hydraulic backwashing every 60 min, with cleaning times ranging from 40 to 60 s. Chemical-enhanced backwashing was carried out every 3 to 7 days. After UF treatment, the effluent firstly entered the NF inlet tank to be pumped into the cartridge filter by a lift pump. The cartridge filter was installed at the outlet of the lift pump to ensure that particulate matter is removed before it enters the NF unit, thereby preventing membrane damage. A dosing system was placed between the lift pump and the cartridge filter. When the residual chlorine content of the water is too high, sodium bisulfite solution is injected into the water to remove the residual chlorine and prevent the NF membrane from being oxidized. During the operation of

the NF unit, a certain amount of scale inhibitor was added between the cartridge filter and the high-pressure pump, which can increase the solubility of calcium and magnesium salt, thereby inhibiting the deposition of scale and reducing the fouling rate of the NF membrane. As for the cleaning water for NF membrane, part of the clean drainage was circulated to the NF system, and the other part was discharged directly. Each kind of NF membrane was tested in the NF unit for 30 days. The inlet flow rate was set at 3.2 m$^3$/h. In order to obtain the water recovery of 90% in this pilot-scale NF system, concentrate was continuously pumped back to the inlet of the first stage with a flow rate of 0.3 m$^3$/h. The frequency of chemical cleaning was once every 24 h.

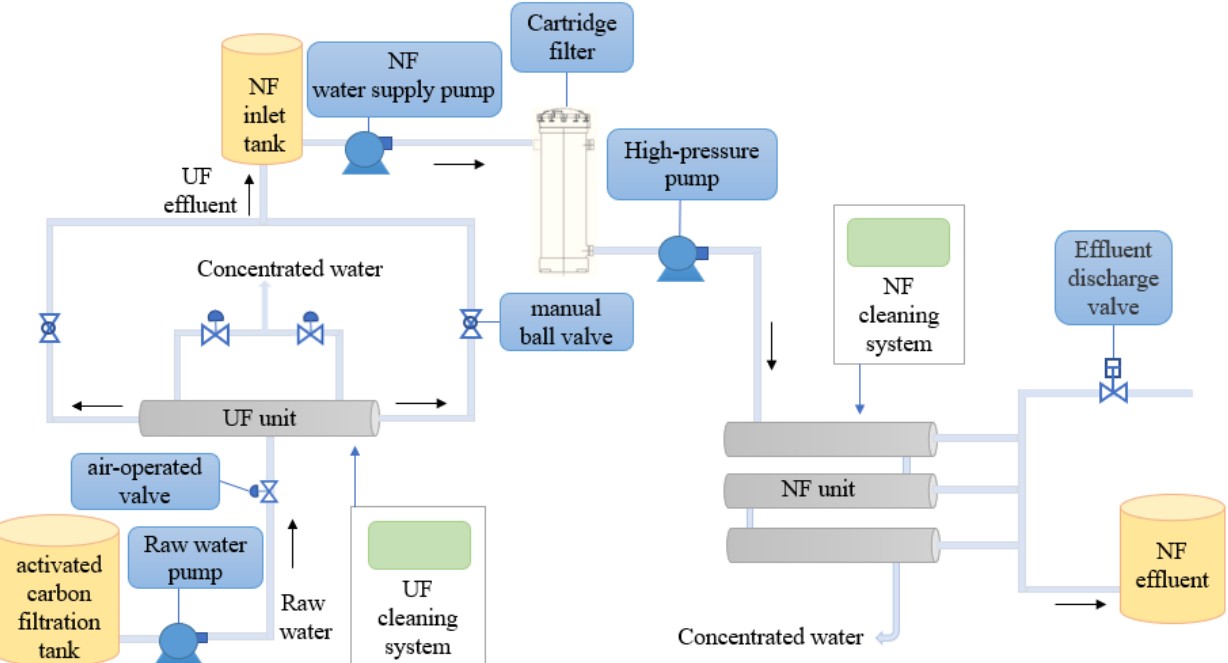

**Figure 1.** Schematic diagram of the pilot scale.

### 2.3. Membrane Information

Compared with NF membranes, the application of UF membranes is more mature and widespread. Therefore, a commonly used commercial UF membrane (dizzer XL 0.9 MB 80W, inge GmbH, Greifenberg, Germany) was selected as the pre-treatment of NF unit.

Three types of commercial NF membranes were selected for this pilot study, namely Dow Filmtec NF270-400 (NF1), VONTRON TAPU-LS (NF2), GE Osmonics-HL8040F 400 (NF3). All these membranes were purchased from GreenTech Co., Ltd., Beijing, China. The membranes were stored at 4 °C and rinsed with purified water before use. According to the information provided by the supplier, all three thin-film membranes (TFMs) are strongly negatively charged and have a proprietary active nano-polymer layer based on polypyperazinamide. Besides, the molecular cut-off for all NF membranes is 100–300 Da. The specifications and performance of the NF membranes are shown in Table 2.

**Table 2.** The specifications and performance of NF membranes applied in this study.

| Specifications and Performance | NF1 | NF2 | NF3 |
|---|---|---|---|
| Effective membrane area (m$^2$) | 37 | 37.2 | 37 |
| MWCO (Da) | 300 | 300 | 300 |
| Contact angle (°) | 26.1 ± 1.0 | 48.8 ± 1.0 | 56.7 ± 1.0 |
| Maximum operating temperature (°C/°F) | 45/113 | 45/115 | 50/122 |
| Maximum operating pressure (bar/psi) | 41/600 | 41.4/600 | 31.03/450 |
| Maximum pressure drop (bar/psi) | 1.0/15 | 1.0/15 | 0.83/12 |

**Table 2.** *Cont.*

| Specifications and Performance | NF1 | NF2 | NF3 |
| --- | --- | --- | --- |
| pH range, continuous operation | 3–10 | 3–10 | 3–9 |
| pH range, short-term cleaning | 1–12 | 2–12 | 2–11 |
| Maximum feed silt density index | SDI5 | SDI5 | SDI5 |
| Maximum feed turbidity (NTU) | 1 | 1 | 1 |
| Free chlorine tolerance (ppm) | <0.1 | <0.1 | <0.1 |
| Salt retention rate | >97% | >97% | >95% |
| Membrane materials | Polyamide | Polyamide | Polyamide |

### 2.4. Pilot-Scale UF+NF System

In this pilot study, a combination of UF and NF apparatuses was adopted, in which the NF unit is composed of three-stage nanofiltration units in series. The on-line instruments of the equipment were well configured, and could run under the PLC automatic control system and automatically complete the operation processes of filtration, cleaning, sewage discharge, etc., as well as record the transmembrane differential pressure, flow rate, power consumption, water consumption (daily inflow and outflow difference) and other data. Operating parameters were set and adjusted manually. The configurations of the main equipment are given in Tables 3 and 4. Figure 2 shows the on-site pictures of UF and NF from this pilot-scale test.

**Table 3.** Equipment parameters of UF unit.

| Equipment | Parameters |
| --- | --- |
| UF membrane element | Membrane area: 77–80 m$^2$. |
| UF inlet tank | Capacity: 1.0 m$^3$. |
| UF backwash tank | Capacity: 1.0 m$^3$. |
| UF cleaning water tank | Capacity: 1.0 m$^3$. |
| Water supply pump | Flow rate: 10 m$^3$/h; Lift: 24 m. |
| Backwash pump | Flow rate: 20 m$^3$/h; Lift: 30 m. |
| Cleaning pump | Flow rate: 4 m$^3$/h; Lift: 20 m. |

**Table 4.** Equipment parameters of NF unit.

| Equipment | Parameters |
| --- | --- |
| NF membrane elements | Membrane area: 37 m$^2$. |
| NF inlet tank | Capacity: 0.5 m$^3$ |
| NF cleaning water tank | Capacity: 0.5 m$^3$ |
| Water supply pump | Flow rate: 8 m$^3$/h; Lift: 30 m. |
| High-pressure pump | Flow rate: 8 m$^3$/h. Lift: 70 m. |
| Cleaning pump | Flow rate: 8 m$^3$/h. Lift: 30 m. |

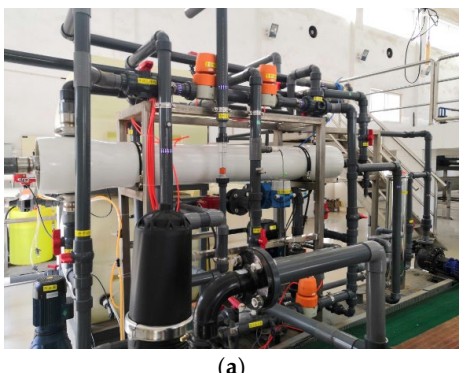 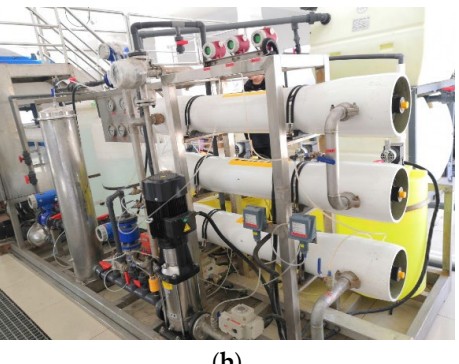

(**a**)　　　　　　　　　　　　　　　　(**b**)

**Figure 2.** On-site pictures of UF and NF. (**a**) UF unit; (**b**) NF unit.

### 2.5. Chemicals

In the UF unit, citric acid solution (2000 mg/L), sodium hydroxide solution (500 mg/L) and sodium hypochlorite solution (200 mg/L) were used as cleaning agents. While in the NF unit, citric acid solution and sodium hydroxide solution were used as cleaning agents. In addition, the UF unit used polyaluminum chloride solution (0.2 mg/L) as a flocculant; the NF unit used sodium bisulfite solution (1 mg/L) to keep the residual chlorine content in the influent within the tolerance range of the NF membrane.

Among them, citric acid and sodium bisulfite were prepared by dissolving analytical grade powder in deionized water. The sodium hydroxide solution and sodium hypochlorite solution were obtained by diluting industrial grade raw material liquid. The polyaluminum chloride solution were prepared by dissolving food grade polyaluminum chloride in deionized water.

Unless otherwise specified, the detection reagents used in the experiment were all analytically pure, and the solutions were all prepared with deionized water.

### 2.6. Analytical Methods

The pH was measured by a portable pH meter (CT-6821, Kedida Electronics Co., LTD, Shenzhen, China). The turbidity was detected by a desktop turbidimeter with an accuracy of 0.1NTU (Hach 2100N, Hach Company, Colorado, CO, USA). The $COD_{Mn}$ was determined using the potassium permanganate method (Chinese SEPA, Beijing, China, 2002). The hardness was detected by EDTA titration (Chinese SEPA, 2002). Chloride was determined by $AgNO_3$ standard solution titration (Chinese SEPA, 2002). The concentration of $NH_3$-N was detected by Nessler's Reagent Spectrophotometry (Chinese SEPA, 2002). Conductivity was measured by a portable conductivity meter (Seven2Go S3, Mettler Toledo Group, Zurich, Switzerland). The silting density index (SDI) value was detected using an SDI measuring instrument (0.45 μm, Simple SDI, SPEARS, Florida, FL, USA).

## 3. Results and Discussion

### 3.1. Pretreatment Effect of UF

The effectiveness of the UF pretreatment was estimated by monitoring the pH, turbidity, and SDI of UF effluent. As shown in Figure 3a, the turbidity of raw water fluctuated between 0.223 and 0.742 NTU, while the average turbidity of UF effluent was only 0.04 NTU. Furthermore, the average turbidity rejection ratio by UF was 88.6 ± 2.9%, with a high of 94.2%. The results indicated that the turbidity of raw water had been significantly reduced by the pre-treatment of UF unit, which created a good environment for the subsequent nanofiltration. The excellent performance of UF pretreatment could also be reflected by $SDI_{15}$. As can be seen in Figure 3b, the maximum $SDI_{15}$ is 2.68, which meets the NF feeding requirement of $SDI_{15} < 5$. In addition, the SDI of UF effluent does not decay rapidly with time.

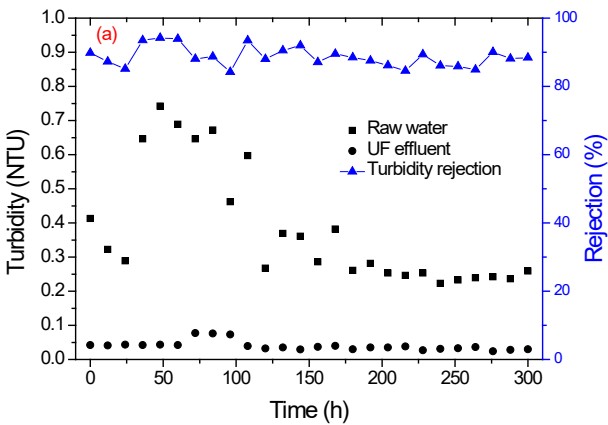 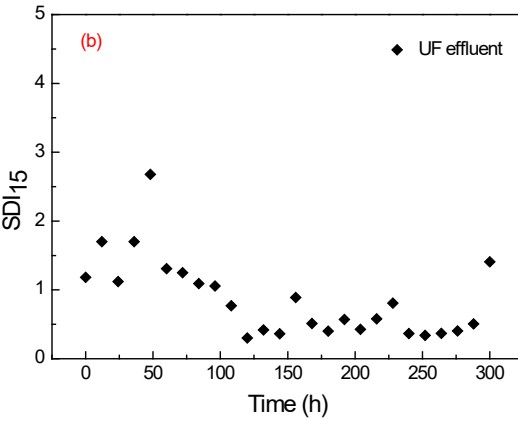

**Figure 3.** (**a**) Turbidity of the raw water and UF effluent with time. (**b**) SDI of the UF effluent with time.

In order to verify the long-term stability of the UF pre-treatment system, the TMP of the UF membrane was detected at a chemical cleaning frequency of 7 days. As shown in Figure 4, after seven rounds of chemical cleaning, TMP did not change significantly and remained around 0.24 bar. After the fifth chemical cleaning, the TMP showed fluctuations, which may be caused by fluctuations in the temperature of the raw water. This is because temperature will affect the viscosity of raw water, and a decrease in the temperature would increase the viscosity of the water, which would lead to the increase in the trans-membrane pressure (TMP) [34]. Therefore, the UF unit could continuously and stably provide qualified filtrate for the NF unit.

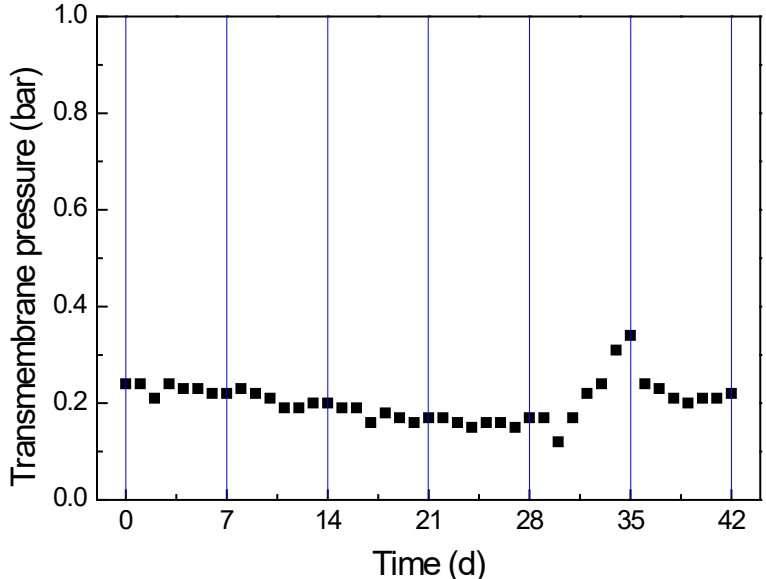

**Figure 4.** TMP as a function of filtration time of water treated by UF. Chemical cleaning interval = 7 d.

### 3.2. Performance of the Pilot-Scale NF System

3.2.1. Performance of NF1 System

In this study, the performance of NF membranes was evaluated by monitoring pH, conductivity, $COD_{Mn}$ and hardness. The NF1 system showed stable and efficient performance during operation. The average flux of NF1 system was $26.1 \pm 0.8$ L/ (m$^2$·h) which was lower than the design value of 27 L/ (m$^2$·h). Moreover, there was no significant change in LMH during the 30-day continuous experiment, indicating that the phenomenon of membrane fouling was not serious in this pilot study. The performance of NF membrane is significantly affected by the pH [34]. During this study, the pH of NF influent was within the permissible range. The average pH value of the effluent was $7.52 \pm 0.04$, which was similar to that of the influent, $7.51 \pm 0.04$. The average pH value of concentrate was obviously higher than the first two, which was $7.67 \pm 0.04$ (Figure 5a). The reason for the small change in influent and effluent pH may be the low removal rate of $HCO_3^-$ by NF1 membrane [35]. The average conductivities of influent and effluent were $557.8 \pm 19.0$ and $396.1 \pm 23.1$ μS/cm, respectively (Figure 5b). The results indicated that the average salt rejection ratio by NF1 was 29.0%. Although the $COD_{Mn}$ value of influent was lower than 5 mg/L, the average $COD_{Mn}$ rejection ratio by NF1 was still up to 67.0%, showing an ideal advanced treatment capacity (Figure 5c). Hardness is also a concern of water plants. The average hardness rejection ratio by NF1 reached $40.8 \pm 2.6\%$ (Figure 5d).

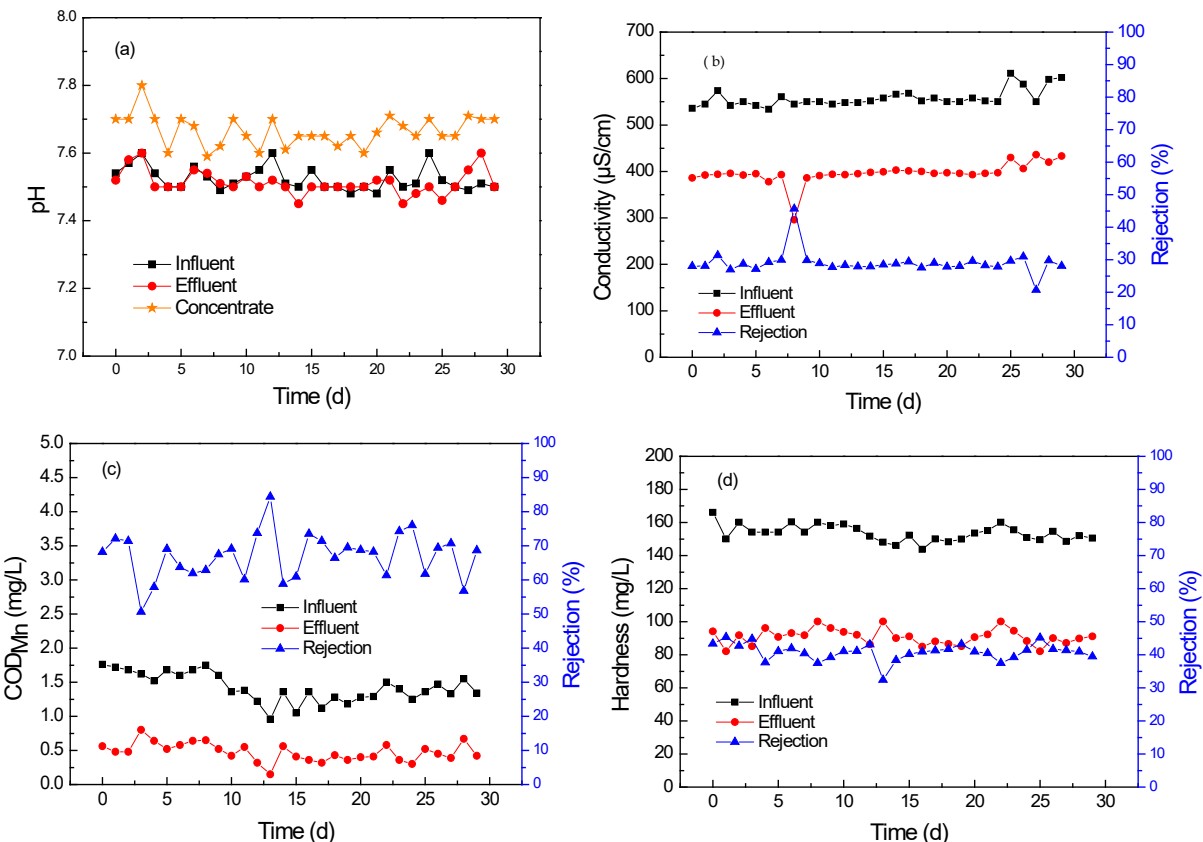

**Figure 5.** Water quality of the pilot-scale NF system with Dow Filmtec NF270-400. (**a**) pH; (**b**) conductivity; (**c**) COD$_{Mn}$; (**d**) hardness.

3.2.2. Performance of NF2 System

The NF2 membrane was tested for the same operation. However, the NF2 system had an average water recovery of 88.2%, slightly below the target of 90%. As shown in Figure 6a, there was a high consistency between the pH values of the effluent and the influent, which were 7.47 ± 0.07 and 7.48 ± 0.05, respectively. In contrast, the pH value of concentrate fluctuated more dramatically, which was 7.65 ± 0.06. The average conductivities of influent and effluent were 510.0 ± 8.8 and 382.8 ± 11.2 μS/cm during the operation, respectively (Figure 6b). The average salt rejection ratio of NF2 was 25.1%. The average COD$_{Mn}$ rejection ratio of NF2 was 75.7 ± 4.4% (Figure 6c). Meanwhile, the average hardness rejection ratio of NF2 was 39.6 ± 1.1%, which was slightly lower than that of NF1 (Figure 6d).

3.2.3. Performance of NF3 System

The NF3 membrane was also tested for the same operation. The average water recovery of NF3 system was 90.1%, which also reached the target value. The pH values of influent and effluent also maintained a high degree of consistency, at 7.48 ± 0.05 and 7.69 ± 0.05, respectively (Figure 7a). As shown in Figure 7b, the average conductivities of influent and effluent were 586.0 ± 15.5 and 424.8 ± 10.6 μS/cm, respectively. This indicates that the average salt rejection ratio of NF3 was 27.5%. The average COD$_{Mn}$ rejection ratio of NF3 was 72.4 ± 4.8% (Figure 7c). Differing from NF1 and NF2, the hardness removal effect of NF3 was significantly better than that of the first two. The average hardness of effluent was only 44.6 mg/L (Figure 7d), which lead to a hardness rejection ratio of 73.2%.

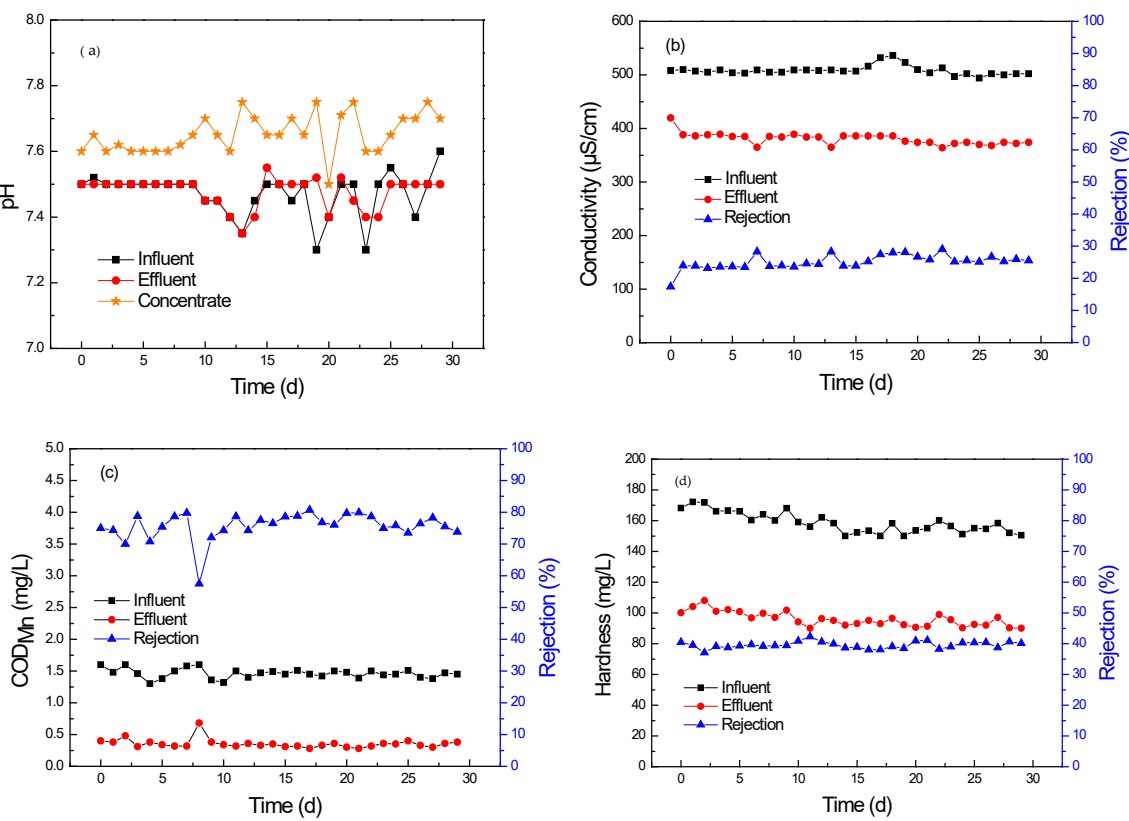

**Figure 6.** Water quality of the pilot-scale NF system with VONTRON TAPU-LS. (**a**) pH; (**b**) conductivity; (**c**) $COD_{Mn}$; (**d**) hardness.

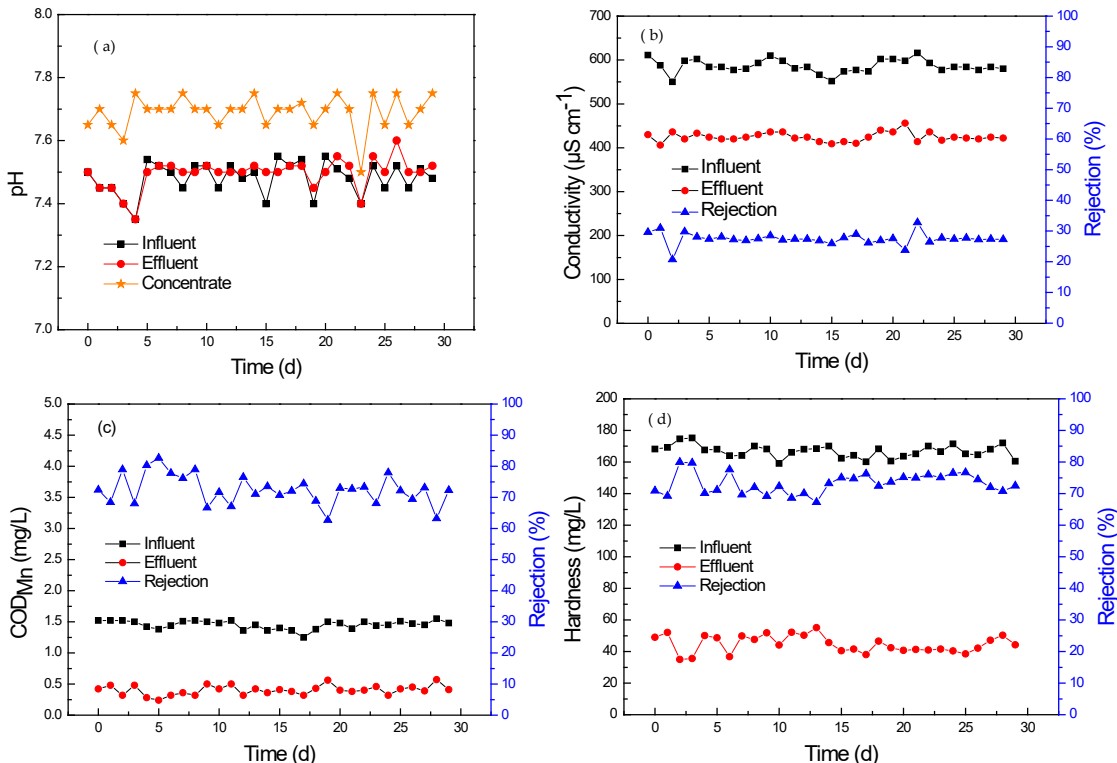

**Figure 7.** Water quality of the pilot-scale NF system with GE Osmonics-HL8040F 400. (**a**) pH; (**b**) conductivity; (**c**) $COD_{Mn}$; (**d**) hardness.

### 3.3. Comprehensive Evaluation Based on Performance and Economy

In this study, the UF membrane and three kinds of NF membrane showed satisfactory stability. The UF membrane acted as the pretreatment unit to ensure that the effluent meets the inlet requirements of the NF membranes. Besides, the pH, conductivity, $COD_{Mn}$ and hardness of the effluent were very stable, which provided a relatively consistent condition for the pilot-scale study of the three NF membranes.

As listed in Table 5, the three NF membranes showed different performance focuses during the experiment, reflecting the different characteristics of the three membranes. The desalting performance of NF1 is the best, and the effluent conductivity can be stabilized below 400 μS/cm. However, it also has the lowest retention rate for organic matter. This is because the desalting performance is mainly determined by the charge effect of the NF membrane, while the retention of neutral non-charged substances (such as lactose, glucose and maltose) is determined by the sieving effect of the NF membrane [27]. Since NF1 has the smallest contact angle, which indicates that its hydrophilicity is the best, its retention performance on hydrophobic organic matter is not as good as the other two NF membranes. NF2 has the best $COD_{Mn}$ rejection performance, but the lowest desalting performance. NF1 and NF2 are similar in performance, but due to the difference in their hydrophilicity, they place different emphasis on salt removal and organic matter removal. NF3 has the best softening performance. Being affected by the Donnan effect, NF membranes have different retention rates for monovalent and divalent ions [36]. The charge numbers of NF1 and NF2 were similar, while NF3 had a stronger charge number, resulting in its effluent conforming to World Health Organization standards for soft water (i.e., with a hardness less than 60 mg/L). Therefore, for advanced treatment in a water plant, NF membranes should be selected according to the quality characteristics of raw water.

**Table 5.** Performance of NF membranes in the pilot-scale system.

| Membrane | Performance | | | |
|---|---|---|---|---|
| | Recovery | Desalting | Purification | Softening |
| NF1 | 90.6% | 29.0% | 67.0% | 40.8% |
| NF2 | 88.2% | 25.1% | 75.7% | 39.6% |
| NF3 | 90.0% | 27.5% | 72.4% | 73.2% |

The operating cost of NF membranes is also an important index to be considered. The electricity consumption and drug consumption of the UF + NF process were evaluated. The electricity consumption includes the power consumption generated by the UF unit, NF unit, water intake pump and drainage pump, etc. Since the other devices except the NF membrane remain consistent in this study, the energy consumption difference in the NF membrane can be seen from the electricity consumption changes of the system. As listed in Table 6, NF1 had the smallest energy consumption, with a cost of 0.165 USD per ton of produced water. The energy consumption of NF2 was the highest, reaching 0.179 USD per ton of produced water. There was no significant difference in drug consumption among the three NF membranes.

**Table 6.** Economic comparison of three NF membranes on pilot scale.

| Membrane | Electricity Consumption [1] kwh/ton Water | Electricity Consumption [2] USD/ton Water | Drug Consumption USD/ton Water | Total USD/ton Water |
|---|---|---|---|---|
| NF1 | 1.29 | 0.138 | 0.027 | 0.165 |
| NF2 | 1.45 | 0.154 | 0.025 | 0.179 |
| NF3 | 1.35 | 0.143 | 0.028 | 0.171 |

[1] The electricity consumption includes the power consumption of the UF unit and NF unit. [2] Converted using the local industrial electricity price.

## 4. Conclusions

In order to adapt to the increasing quality of tap water standards, water plants are looking for cost-effective advanced treatment technologies. In this study, the effluent from an advanced carbon filter was used as raw water to investigate the advanced treatment effect of the pilot-scale system of UF + NF. The results showed that the UF membrane, as the pretreatment unit of the NF unit, could reduce the turbidity of raw water by 88.6%, so that the SDI of the effluent is less than 3, thereby meeting the influent requirements of the NF membrane. In addition, the flux of the UF membrane did not attenuate significantly during operation. The three NF membranes combined with the UF unit showed excellent performance and good stability. With a water recovery of 90%, the conductivity, $COD_{Mn}$ and hardness of the NF effluent are significantly improved. Meanwhile, there was no significant change in the pH of both the effluent and influent. In this study, three commercial NF membranes showed different performance advantages. Among them, NF1 had the best desalting performance, NF2 had the highest retention rate of organic matter, and NF3 had a significantly advanced softening performance. In terms of economics, NF1 performed best, costing 0.165 USD per ton of produced water. Therefore, the UF + NF process is an ideal advanced treatment technology for water plants. The results of this study have important reference value for the upgrading of water plants worldwide.

**Author Contributions:** Conceptualization, L.Z.; methodology, L.Z.; software, F.C.; validation, F.C. and F.B.; formal analysis, F.Y. and J.T.; investigation, F.C.; resources, Z.Y. and D.L.; data curation, N.G.; writing—original draft preparation, F.C.; writing—review and editing, L.C. and L.Z.; supervision, L.Z. All authors have read and agreed to the published version of the manuscript.

**Funding:** The research was funded by the Key Laboratory for Technology in Rural Water Management of Zhejiang Province, grant number ZJWEU-RWM-202101, and the Zhejiang Provincial Natural Science Foundation of China, grant number LZJWZ22C030001.

**Institutional Review Board Statement:** Not applicable.

**Informed Consent Statement:** Not applicable.

**Data Availability Statement:** The data used to support the findings of this study are currently under embargo while the research findings are commercialized. Requests for data made 6 months after publication of this article will be considered for uploading to the repository of FIGSHARE.

**Acknowledgments:** The authors appreciate the support of the Key Laboratory for Technology in Rural Water Management of Zhejiang Province, and the Zhejiang Provincial Natural Science Foundation of China.

**Conflicts of Interest:** The authors declare no conflict of interest.

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
