# Peer review of "A Pilot-Scale Nanofiltration–Ultrafiltration Integrated System for Advanced Drinking Water Treatment: Process Performance and Economic Analysis"

_processes, doi:10.3390/pr11051300_

Round 1
Reviewer 1 Report
11) The authors must explain common water contaminants in the manuscript.
22) Please amend the existed discrepancies in the references and observe similar formatting style.
3) Please bring commonly-employed membranes for water treatment.
4) What are the advantages of ultrafiltration membranes compared to nanofiltration membrane.
25) Introduction is weak. Please enrich some more updated references with explanations. I suggest the references https://doi.org/10.3390/membranes12040429; https://doi.org/10.3390/membranes12040429
6) Please increase the quality of Figure 1.
77) English of manuscript needs correction.
8) Some parts of “results and discussion” section must be explained more.
Reviewer 2 Report
• What is the novelty of this study? It is very common to have a complete NF system coupling UF as a pretreatment process in the water treatment technology. Please highlight it.
• Typically, NF has limitations in completely removing single ions such as NaCI presence in the solution, please explain what makes the NF membranes adopted in this pilot scale study outperformed any common NF? What are their special characteristics?
• What is the source of water? It should be mentioned in the manuscripts.
• If it is surface water, it is a high chance that the membrane water treatment would be experiencing biofouling. How is the system going to cope with biofouling since no sodium hypochlorite (NaClO) is being incorporated into the filtration system as the cleaning agent for NF? I really doubt the functionality of the system.
• From Table 1, why only selected paramaters been investigated? For drinking water quality ruled by WHO, there are more than what has been reported in Table 1. Authors should consider more parameters to reflect the usage of potable water.
• In Section 2.4, it is stated that a 3-stage NF in series is configured. However, a more detailed system design configuration should be provided which includes feed pressure, total nos of element in 1 stage, nos of pass, etc. How long is the duration of the experiment for the pilot scale? Continuous or intermittent mode? Authors should cite the following paper “Recycling of oleochemical wastewater for boiler feed water using reverse osmosis membranes — A case study Desalination, 271, Issues 1–3, Pages 178-186”, which provided a detailed discussion on materials&methods and discussion pertaining to the case study of water recycling system using membrane technology to enhance the presentation of the manuscript.
• The authors mentioned about chemical cleaning using citric acid solution and NaOH as cleaning agents for NF. Which concentrations of citric acid solution and NaOH were used for system cleaning? Does the pH of cleaning solution meet the pH range required by the membrane manufacturers?
• How was the performance of UF in terms of rejection? Why was only the turbidity being presented?
Reviewer 3 Report
The authors presented a paper entitled “Nanofiltration for advanced drinking water treatment in pilot scale: Membrane characteristics and process performance”.
Bellow, some comments which should be reported.
1. Introduction
In the introduction authors mentioned most of the membrane processes in water treatment. Maybe some words can be added regarding the deionization process in water treatment or hybrid processes.
2. Material and Methods
Line 102: Table 1. Insist of Indicator can be Parameter.
Line 106: UF and NF equipment should be described.
Line 139: The table should be in the format, and looks like a copied specification.
Line 154: Table 4. Insist of Main equipment can be Equipment.
The authors mentioned membrane characteristics. First of all, it promised membrane stability or maybe a scaling investigation. From my point, there are missing methods and results of UF/NF membranes.
Reviewer 4 Report
The present manuscript reports on the “Nanofiltration for advanced drinking water treatment in pilot scale: Membrane characteristics and process performance”. The work is of some interest. Thus, in my opinion, the manuscript in its present form cannot be considered for publication.
Following are some of the comments/suggestions which will be useful to the authors.
1. First of all, there are many previous works published for purification of water by some other methods such as catalysis, and adsorption. The authors seem deliberately avoid those papers. The author should cite such articles for comparison of this work along with others. There are some articles for this purpose. Cite these articles some proper place. doi.org/10.1016/j.jece.2023.109270; doi.org/10.1016/j.molliq.2023.121346; doi.org/10.1515/zpch-2018-1182; doi.org/10.1039/C5RA05785J
2. Title of this article should be improved.
3. A table of comparison is required. The results of this article should be compared with previous in this table.
4. Write the conditions for performance of purification such as temperature, pH, and pressure etc. in fig. 3-7. These conditions take part important role in purification process.
5. Improve discussion portion in results and discussion.
6.Add more citations in results and discussion portion.
Round 2
Reviewer 1 Report
the revised version of manuscript is acceptable.
Author Response
The author would like to express our gratitude to the reviewer for your positive and encouraging comments on our manuscript.
Reviewer 2 Report
All the highlighted comments have been sufficiently addressed
Author Response

(The authors gave the same response as above.)

Reviewer 3 Report
Bellow, some comments which should be reported.
Title: should be (System).
maybe Membrane Characteristics are not important in the title.
Please check the abbreviations:
Line 20: COD
Line 69: NTU
Line 220: Trans-membrane pressure (TMP), should be TMP before that you mentioned in 226.
Line 237: LMH
